# Algebraic graph-assisted bidirectional transformers for molecular property prediction

Dong Chen [1,2], Kaifu Gao[2], Duc Duy Nguyen[3], Xin Chen[1], Yi Jiang[1], Guo-Wei Wei [2,4,5✉] & Feng Pan [1✉]

The ability of molecular property prediction is of great significance to drug discovery, human health, and environmental protection. Despite considerable efforts, quantitative prediction of various molecular properties remains a challenge. Although some machine learning models, such as bidirectional encoder from transformer, can incorporate massive unlabeled molecular data into molecular representations via a self-supervised learning strategy, it neglects three-dimensional (3D) stereochemical information. Algebraic graph, specifically, element-specific multiscale weighted colored algebraic graph, embeds complementary 3D molecular information into graph invariants. We propose an algebraic graph-assisted bidirectional transformer (AGBT) framework by fusing representations generated by algebraic graph and bidirectional transformer, as well as a variety of machine learning algorithms, including decision trees, multitask learning, and deep neural networks. We validate the proposed AGBT framework on eight molecular datasets, involving quantitative toxicity, physical chemistry, and physiology datasets. Extensive numerical experiments have shown that AGBT is a state-of-the-art framework for molecular property prediction.

[1] School of Advanced Materials, Peking University, Shenzhen Graduate School, Shenzhen, China. [2] Department of Mathematics, Michigan State University, East Lansing, MI, USA. [3] Department of Mathematics, University of Kentucky, Lexington, KY, USA. [4] Department of Electrical and Computer Engineering, Michigan State University, East Lansing, MI, USA. [5] Department of Biochemistry and Molecular Biology, Michigan State University, East Lansing, MI, USA. ✉email: weig@msu.edu; panfeng@pkusz.edu.cn

The fact that there is no specific and effective drug for coronavirus disease 2019 (COVID-19) 1 year after the outbreak reminds us that drug discovery remains a grand challenge. Rational drug discovery involves a long list of molecular properties, including binding affinity, toxicity, partition coefficient, solubility, pharmacokinetics, pharmacodynamics, etc[1]. Experimental determination of molecular properties is very time-consuming and expensive. In addition, experimental testing involving animals or humans is subject to serious ethical concerns. Therefore, various computer-aided or in silico approaches have become highly attractive because they can produce quick results without seriously sacrificing accuracy in many cases[2]. One of the most popular approaches is the quantitative structure-activity relationship analysis. It assumes that similar molecules have similar bioactivities and physicochemical properties[3].

Recently, machine learning (ML), including deep learning (DL), has emerged as a powerful approach for data-driven discovery in molecular science. For example, graph convolutional networks (GCNs)[4–6], convolutional neural networks (CNNs)[7], and recurrent neural networks (RNNs)[8], have become popular for drug discovery and molecular analysis[8–10]. Generative adversarial networks (GANs)[11] combined with some machine learning strategies, such as supervised learning and reinforcement learning, have also been applied to the generation of novel molecules and drug design[12]. However, DL methods require large datasets to determine their large number of weights and might not be competitive for small datasets[13].

Although DL methods, particularly CNN and GANs, can automatically extract features from simple data, such as images and/or texts, the performance of ML and DL methods for molecules, particularly macromolecules, crucially depends on the molecular descriptors or molecular representations due to their intricate structural complexity[14]. Earlier molecular descriptors are designed as the profiles or fingerprints of interpretable physical properties in a bit string format[15]. Various fingerprints have been developed in the past few decades[16,17]. There are four main categories of two-dimensional (2D) fingerprints[17], namely substructure key-based fingerprints[18], topological or path-based fingerprints[19], circular fingerprints[16], and pharmacophore fingerprints[20]. However, 2D fingerprints lack three-dimensional (3D) structural information of molecules, especially stereochemical descriptions.

To deal with the aforementioned problems, 3D-structure-based fingerprints have been developed to capture 3D patterns of molecules[21]. However, the molecular structural complexity and high dimensionality are the major obstacles in designing efficient 3D fingerprints[14]. Recently, a variety of 3D molecular representations based on advanced mathematics, including algebraic topology[7,22], differential geometry[23], and algebraic graph[24] have been proposed to simplify the structural complexity and reduce the dimensionality of molecules and biomolecules[14,25]. These methods have had tremendous success in protein classification, and the predictions of solubility, solvation-free energies, toxicity, partition coefficients, protein folding stability changes upon mutation, and Drug Design Data Resource (D3R) Grand Challenges[14,26], a worldwide competition series in computer-aided drug design. However, this approach depends on the availability of reliable 3D molecular structures.

Alternatively, a self-supervised learning (SSL) strategy can be used to pre-train an encoder model that can produce latent space vectors as molecular representations without 3D molecular structures. The unlabeled data is used in the SSL strategy, but unlike unsupervised learning, the data input to the model is partially masked, and then the model is trained to predict the masked part in the training process, where the originally masked data can be used as labels. This strategy allows a large amount of unlabeled data to be utilized. The initial development of SSL was due to the need for natural language processing (NLP)[27,28]. For example, bidirectional encoder representations from transformers (BERTs) are designed to pre-train deep bidirectional transformer representations from unlabeled texts[27]. The techniques developed in understanding sequential words and sentences in NLP have been used for understanding the fundamental constitutional principles of molecules expressed as a simplified molecular-input line-entry system (SMILES)[29]. Unlabeled SMILES strings can be considered as text-based chemical sentences and are used as inputs for SSL pre-training[28,30]. It is worth noting that the availability of large public chemical databases such as ZINC[31] and ChEMBL[32] makes SSL a viable option for molecular representation generation. However, latent-space representations ignore much stereochemical information, such as the dihedral angle[33] and chirality[34]. In addition, latent-space representations lack specific physical and chemical knowledge about task-specific properties. For example, van der Waals interactions can play a greater role than the covalent interactions in many drug-related properties[35], and need to be considered in the description of these properties.

In this work, we introduce algebraic graph-assisted bidirectional transformer (AGBT) to construct molecular representations via combining the advantages of 3D element-specific weighted colored algebraic graphs and deep bidirectional transformers. The element-specific weighted colored algebraic graphs generate intrinsically low-dimensional molecular representations, called algebraic graph-based fingerprints (AG-FPs), that significantly reduce the molecular structural complexity while retaining essentially physical/chemical information and physical insight[24]. Deep bidirectional transformer (DBT) utilizes an SSL-based pre-training process to learn fundamental constitutional principles from massive unlabeled SMILES data and a fine-tuning procedure to further train the model with task-specific data. The resulting molecular fingerprints, called bidirectional transformer-based fingerprints (BT-FPs), are latent-space vectors of the DBT. The proposed AGBT model is applied to eight benchmark molecular datasets involving quantitative toxicity and partition coefficient[2,13,36,37]. Extensive validation and comparison suggest that the proposed AGBT model gives rise to some of the best predictions of molecular properties.

## Results

In this section, we present the proposed AGBT model and its results for molecular prediction on eight datasets, i.e., LD50, IGC50, LC50, LC50DM, partition coefficient, FreeSolv, Lipophilicity, and BBBP datasets. Supplementary Table 1 lists the basic information of these datasets and the CheMBL[32] dataset was used in the pre-training. More descriptions of the datasets can be found in Supplementary Note 1.

**Algebraic graph-assisted deep bidirectional transformer (AGBT).** As shown in Fig. 1, the proposed AGBT consists of four major modules: AG-FP generator (i.e., the blue rectangles), BT-FP generator (i.e., the orange rectangles), random forest (RF)-based feature-fusion module (i.e., the green rectangle), and downstream machine learning module (i.e., the pink rectangle). For the graph fingerprint generation, we use element-specific multiscale weighted colored algebraic graphs to encode the chemical and physical interactions into graph invariants and capture 3D molecular structural information. The BT-FPs have created in two steps: an SSL-based pre-training step with massive unlabeled input data and a task-specific fine-tuning step. The task-specific fine-tuning step can be executed in two ways. The first way is merely to adopt the same SSL procedure to fine-tune the model

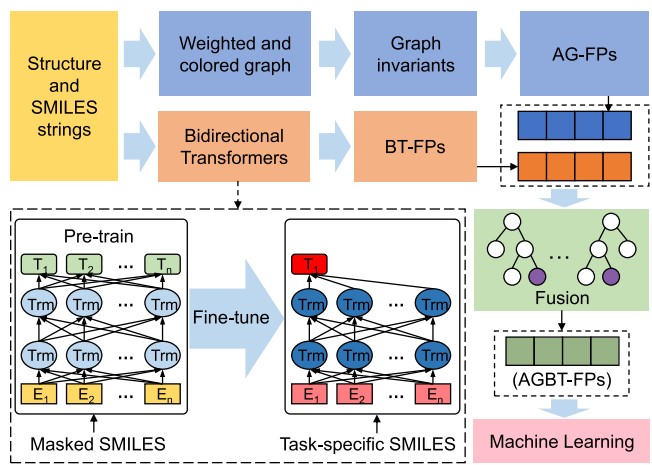

**Fig. 1 Illustration of AGBT model.** For a given molecular structure and its SMILES strings, AG-FPs are generated from element-specific algebraic subgraphs module and BT-FPs are generated from a deep bidirectional transformer module, as shown inside the dashed rectangle, which contains the pre-training and fine-tuning processes, and then finally completes the feature extraction using task-specific SMILES as input. Then the random forest algorithm is used to fuse, rank, and select optimal fingerprints (AGBT-FPs) for machine learning.

with task-specific data and generate their BT-FPs. The other way is to utilize labels in task-specific data via a SL procedure to fine-tuning model and generate latent-space vectors of task-specific data, denoted as $BT_s$-FPs (i.e, the orange vector). The random forest algorithm is used to rank the importance of fused AG-FP and BT-PF features and select an optimal set of AGBT-FPs of a fixed number of components. The downstream machine learning algorithms are fed with optimal features to achieve the best performance on four benchmark toxicity datasets.

We carry out our final predictions by using some standard machine learning algorithms, namely, gradient boosted decision tree (GBDT), random forest (RF), and deep neural networks (DNNs), including single-task DNN (ST-DNN, Supplementary Fig. 10a) and multitask DNN (MT-DNN, Supplementary Fig. 10b). Our training follows the traditional pipeline[38]. To eliminate systematic errors in the machine learning models, for each machine learning algorithm, the consensus of the predicted values from 20 different models (generated with different random seeds) was taken for each molecule. Note that the consensus value here refers to the average of the predicted results from different models for each molecule of each specific training-test splitting. In this work, the squared Pearson correlation coefficient ($R^2$), root-mean-square error (RMSE), and mean absolute error (MAE) are used to assess the performance of the regression task, while the classification accuracy and the area under the receiver operating characteristic convex hull (AUC-ROC) are used to evaluate the performance of classification model. All definitions are given in Supplementary Note 2. Further details on our AGBT model are given in the "Method" Section and the parameters set is provided in Supplementary Note 3.

**Toxicity prediction.** Toxicity, a critical issue to consider in drug lead optimization, measures the degree to which a chemical compound can affect an organism adversely[2]. Indeed, toxicity and side effects are responsible for more than half of drug candidate failures on their path to the market[39]. The LC50DM set refers to the concentration of test chemicals in the water in milligrams per liter that cause 50% Daphnia Magna to die after 48 h. Its size is the smallest among the four datasets. Among its 353 molecules,

283 are used as a training set and the rest 70 as a test set[2]. The small size leads to difficulties in training a good prediction model. The overfitting issue poses a challenge to traditional machine learning methods if a large number of descriptors is used. In this work, the MT-DNN is applied to extract information from data sets that share certain statistical distributions, which can effectively improve the predictive ability of models and avoided overfitting on the small datasets[2,13].

Based on the AGBT framework, we fuse AG-FPs and $BT_s$-FPs, i.e., BT-FPs with a supervised fine-tuning procedure for task-specific data. The best performance is obtained by the MT-DNN model, which $R^2 = 0.830$ and RMSE = 0.743. As shown in Fig. 2b, our model yields the best result, which is over 13% better than the previous best score of $R^2 = 0.733$.

The IGC50 set is the second-largest toxicity set and its toxicity values range from 0.334 $-\log_{10}$ mol/L to 6.36 $-\log_{10}$ mol/L[2]. As shown in Fig. 2a, the $R^2$s from different methods fluctuate from 0.274 to 0.810 Karim et al.[40] also studied IGC50 dataset, but their training set and test set are different from those of others[2] and thus their results cannot be included in the present comparison. For our method, the $R^2$ of MT-DNN with AGBT-FP is 0.842, which exceeds that of all existing methods on the dataset IGC50.

The oral rat LD50 set measures the number of chemicals that can kill half of the rats when orally ingested[36,37,41]. This dataset is the largest set among the four sets with as many as 7413 compounds. However, a large range of values in this set makes it relatively difficult to predict[42]. Gao et al.[17] studied this problem using many 2D molecular fingerprints and various machine learning methods, which include GBDT, ST-DNN, and MT-DNN. However, the prediction accuracy of the LD50 data set was not improved much. As shown in Table 1 (the complete comparison in Supplementary Table 6), the $R^2$ values for all existing methods range from 0.392 to 0.643. In our case, our method can achieve $R^2$ 0.671 and RMSE 0.554 log(mol/L), which are better than those methods.

LC50 dataset reports the concentration of test chemicals in water by milligrams per liter that cause 50% of fathead minnows\ to die after 96 h[41]. Wu et al.[2] used physical information including energy, surface energy, electric charge, and so on to construct molecular descriptors. These physical properties are related to molecular toxicity, achieving the prediction accuracy of $R^2$ 0.771. In this work, our AGBT-FPs with MT-DNN deliver the best $R^2$ of 0.776. We also test the performance of our BT-FPs, which achieve $R^2$ 0.783 with MT-DNN. As listed in Table 1, our model outperforms all other existing methods.

**Partition coefficient prediction.** Partition coefficient denoted $P$, derived from the ratio of the concentration of a mixture of two mutually insoluble solvents (octanol and water in these data) at equilibrium, measures the drug relevance of the compound as well as its hydrophobicity to the human bodies. The logarithm of this coefficient is denoted as $\log P$[43]. The training set used for $\log P$ prediction includes 8199 molecules[44]. A set of 406 molecules approved by the Food and Drug Administration (FDA) is used as organic drugs were used as the test set[44] and its $\log P$ values range from $-3.1$ to 7.57. The comparison of different prediction methods for FDA molecular data set is listed in Table 1 and Supplementary Table 6. It should be mentioned that the ALOGPS model established by Tetko et al.[45] can also be used in $\log P$ prediction, however, there is no guarantee that the training set of ALOGPS are independent of the test set and thus its result is not included in the comparison. As we can see from Table 1, our $AGBT_s$-FPs with STDNN model produce the best $R^2$ of 0.905. The predicted result of $AGBT_s$-FPs with STDNN model for FDA data set are shown in Supplementary Fig. 11c.

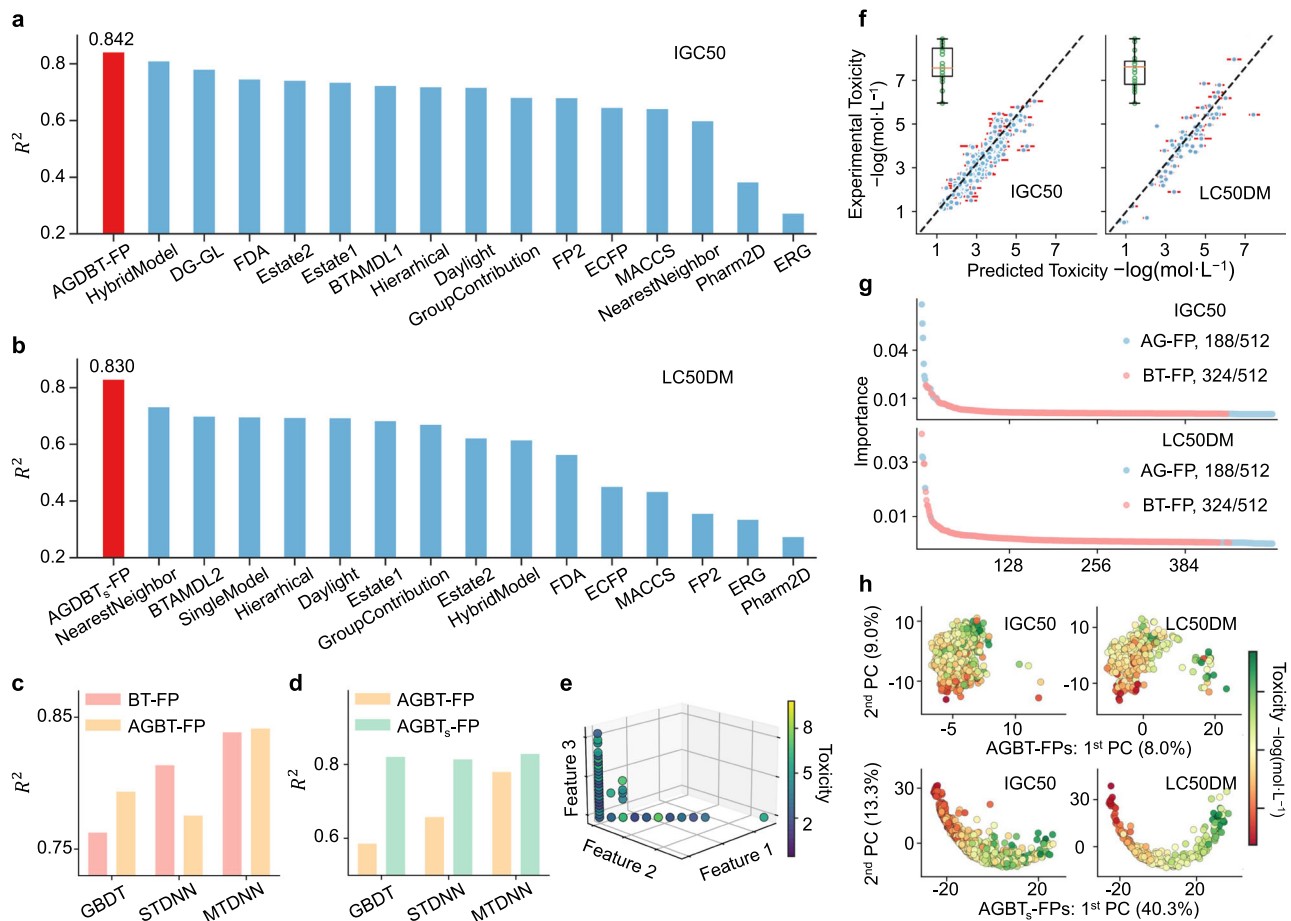

**Fig. 2 Results from AGBT framework and feature analysis. a, b** Illustrate the comparison of the $R^2$ by various methods for the IGC50 set and the LC50DM set, respectively. $AGBT_s$-FP means a supervised fine-tuning process is applied to AGBT-FP. The other results were taken from refs. [2,13,17,23,40,41]. **c** The bar charts illustrate the $R^2$ of AGBT-FPs and BT-FPs with three machine learning algorithms for the IGC50 dataset. **d** The bar charts illustrate the consensus $R^2$ of AGBT-FPs and $AGBT_s$-FPs with three machine learning algorithms for the LC50DM dataset. **e** Visualization of the LD50 set. The axes are the top three important features of AGBT-FPs. **f** Predicted results of AGBT-FPs with MT-DNN model for the IGC50 set(left) and the LC50DM set(right), respectively. The box plots statistic $R^2$ values for $n = 358$ (left), and 70 (right) independent samples examined over 20 independent machine learning experiments, and the detailed statistic values are listed in Supplementary Table 5. **g** The AGBT-FPs of the IGC50 and LC50DM datasets were ranked by their feature importance. For both datasets, 188/512 of the AGBT features are from AG-FPs and the remaining 348/512 are from BT-FPs. **h** Based on AGBT-FPs and $AGBT_s$-FPs, the variance ratios in the first two components from the principal component analysis (PCA) are used to visualize the IGC50 and LC50DM datasets.

**FreeSolv and lipophilicity prediction**. Solvation-free energy and lipophilicity are basic physical and chemical properties for understanding how molecules interact with solvents. In this work, FreeSolv and Lipophilicity datasets are derived from MoleculeNet[9], which is a benchmark for molecular property prediction. There are 643 samples and 4200 samples for FreeSolv and Lipophilicity datasets, respectively. For comparison, the datasets were split into train, validation, test sets with the ratio of 8:1:1, which follows the same procedure as MoleculeNet[9]. We set different random seeds and follow the same procedure ten times to obtain ten different data splitting. To eliminate systematic errors in downstream machine learning models and to better compare molecular descriptors, for each machine learning algorithm, the consensus of the predicted values from 20 models of different random seeds for each data-splitting was taken for each molecule. And the final score for the dataset is the average score over ten different data-splittings. As shown in Table 1, using the STDNN algorithm, our method obtains the best RMSE of 0.994 on FreeSolv, which is better than the best results reported by Chemprop[46] (RMSE = 1.075), MMNB[47] (RMSE = 1.155), and MoleculeNet[9] (RMSE$_{GraphConv}$ = 1.15). For the Lipophilicity

dataset, using the RF algorithm, the RF method, our descriptors obtains the best RMSE of $0.573 \pm 0.026$, a result that is close to the best result reported by Chemprop[46] (RMSE$_{GraphCov}$ = 0.555). A complete result with multiple evaluation metrics is available in Supplementary Table 7.

**Classification task for binary labels of blood–brain barrier penetration (BBBP)**. The BBBP dataset contains 2042 small molecules and original from a study on the modeling and prediction of barrier permeability[48]. The binary labels for compound permeability properties are used in this study. For a better comparison, we follow the same scaffold splitting method described in MoleculeNet[9]. The dataset was split into training, validation, and the test set follow the ratio of 8/1/1. Table 1 reports the best achieved AUC-ROC of our method is 0.763, which is better than results reported in Chemprop[46] (AUC-ROC = 0.738), MMNB[47] (AUC-ROC = 0.739), and MoleculeNet[9] (AUC-ROC$_{ECFP}$ = 0.671).

It is worth noting that to validate the performance of our AGBT framework, for downstream machine learning models, we

**Table 1 Comparison of the best-achieved performance with the reported score on six datasets.**

| LD50 | | LC50 | | FDA | |
|---|---|---|---|---|---|
| Method | $R^2$ | Method | $R^2$ | Method | $R^2$ |
| Ours | **0.671** | Ours | **0.783** | Ours | **0.905** |
| MACCS[17] | 0.643 | BTAMDL2[13] | 0.750 | ESTD-1[43] | 0.893 |
| FP2[17] | 0.631 | ESTDS[2] | 0.745 | Estate2[17] | 0.893 |
| HybridModel[40] | 0.629 | Daylight-MTDNN[17] | 0.724 | XLOGP3[44] | 0.872 |

| FreeSolv | | Lipophilicity | | BBBP | |
|---|---|---|---|---|---|
| Method | RMSE | Method | RMSE | Method | AUC-ROC |
| Ours | **0.994** | Ours | 0.570 | Ours | **0.763** |
| MMNB[47] | 1.155 | MMNB[47] | 0.625 | MMNB[47] | 0.739 |
| Chemprop[46] | 1.075 | Chemprop[46] | **0.555** | Chemprop[46] | 0.738 |
| GraphConv[9] | 1.15 | GraphConv[9] | 0.715 | ECFP[9] | 0.671 |

The best result for each data set has been bolded. Comparison of the $R^2$ for the rest datasets, IGC50, and LC50DM, are shown in Fig. 2 and Supplementary Table 6. A complete table with multiple evaluation metrics for all eight datasets is available in Supplementary Table 7.

did not over-tune the parameters to optimize our results in this study. For all the above-mentioned datasets, the same set of parameters is taken for each type of descriptors in our model. The details of the parameter settings can be found in Supplementary Note 3. Our method exhibits state-of-the-art results in seven out of eight above-mentioned datasets. This illustrates the stable and robust performance of our AGBT framework and its applicability to a wide range of molecular prediction tasks.

## Discussion

In this section, we discuss how the AGBT model brings insights to molecular property predictions, as well as the enhancement that algebraic graph-based fingerprints and deep bidirectional transformers-based fingerprints give rise to our proposed AGBT method.

**Impact of algebraic graph descriptor**. Pre-trained on a large number of molecules, deep SSL-based molecular fingerprints could achieve high accuracy. Many deep learning-based molecular fingerprints have shown a better performance than conventional fingerprints. However, deep learning fingerprints, including our BT-FPs, are prone to the loss of molecular stereochemical information. This lack of information often makes "activity cliff" a nuisance. An example in Supplementary Fig. 16 demonstrates this drawback. Therefore, we propose the use of algebraic graph theory in association with our AGBT framework to retain stereochemical and physical information and enhance the performance of original BT-FPs. Moreover, in this work, we set the total dimension of molecular fingerprints after feature fusion to 512, and thus we only need to optimize one neural network architecture. Our AGBT model is an efficient framework for molecular property predictions.

Figure 2f shows the best prediction performance on the IGC50 and LC50DM datasets using the AGBT framework, namely, $R^2 = 0.842$ on IGC50 and $R^2 = 0.830$ on LC50DM. The orange bar at each point is the deviation of predicted toxicity with 20 models (with different random seed). For each model, $R^2$ was calculated and the distribution of $R^2$ is shown in subfigures. The performance on the LD50 and LC50 datasets is shown in Supplementary Fig. 11b. For LD50, IGC50, and LC50DM datasets, the best prediction results are obtained by the algebraic graph-assisted with MTDNN. For the IGC50 dataset, the $R^2$ of

the toxicity predictions from the three machine learning algorithms, i.e., GBDT, ST-DNN, and MT-DNN, are shown in the bar plot of Fig. 2c. It is obvious that, for the IGC50 dataset, AGBT-FP performs better than BT-FP with GBDT and MT-DNN, while it shows the opposite result on the STDNN. It is mainly because that AG-FPs and BT-FPs are produced from two different molecular fingerprint generators and have the dimensions of 1800 and 512, respectively. The fused molecular fingerprints, AGBT-FPs, return 512 components with heterogeneous information from AG-FPs and BT-FPs and tend to cause some anomalies in the STDNN method.

For the IGC50 dataset, 1434 molecular structures were used to train the AGBT model, leading to fluctuation in prediction Fig. 2f. Similar situations are found in the LD50 dataset and the LC50DM dataset, as shown in Supplementary Fig. 11. For the LC50 dataset, the best result is obtained with BT-FPs, but the result of AGBT-FPs also reaches $R^2$ 0.776, exceeding the other reported methods. As shown in Table 2 and Supplementary Table 8, for FreeSolv and Lipophilicity datasets, the best results are all generated by using fused descriptors, which illustrate that algebraic graph does have an important impact on the molecular property prediction. In Supplementary Table 8, the standard deviation on $R^2$, RMSE, and MAE of FreeSolv and Lipophilicity's prediction also show that $AGBT_s$-FP can obtain the most stable performance in most cases (5/6). Therefore, the fusion of AG-FPs and BT-FPs improves the accuracy and stability of predictions for most datasets. Mathematically based molecular descriptors can complement data-driven potential spatial descriptors.

**Predictive power of fine-tuning strategies**. In this work, we develop two strategies in the fine-tuning stage: SSL and SL with task-specific data. It is found that SSL strategy (See Supplementary Fig. 3 performs better on LD50, IGC50, and LC50 data sets, as shown in Fig. 2f and Supplementary Fig. 11, while SL strategy with task-specific data (See Supplementary Fig. 4 is the best for LC50DM dataset. The LC50DM dataset is the smallest set with only 283 molecules in its training set. Conventional methods cannot capture enough information from such a small dataset to achieve satisfactory results. In the AGBT model, the pre-training strategy with a bidirectional transformer enables the model to acquire a general knowledge of molecules. During the fine-tuning phase, we further feed the model with four toxicity datasets with labels, and the labeled data guide the model to specifically extract toxin-related information from all the training data. Then we complement fine-tuning fingerprints with algebraic graph descriptors to ultimately enhance the robustness of the AGBT model and improve the performance on the LC50DM set ($R^2 = 0.830$, RMSE = 0.743).

Figure 2d shows the performance of AGBT-FPs and $AGBT_s$-FPs on the LC50DM dataset using three advanced machine learning methods. The bar charts show the $R^2$ of prediction results with three machine learning algorithms. This figure shows that $AGBT_s$-FPs have an excellent performance with all three machine learning algorithms, with $R^2$ values being 0.822 (GBDT), 0.815 (ST-DNN), and 0.830 (MT-DNN), respectively. This indicates that $AGBT_s$-FPs can capture general toxin-related information during the sequential fine-tuning process. There is no significant difference among the three predictions based on GBDT, ST-DNN, and MT-DNN. In contrast, AGBT-FPs are derived from the model after self-supervised training. Their pre-training and fine-tuning processes do not involve any labeled data. The resulting prediction accuracies with GBDT and ST-DNN are quite low with $R^2$ being 0.587 and 0.659, respectively. Through the MT-DNN model, the performance of AGBT-FPs can be improved from $R^2$ 0.587 to 0.781.

**Table 2 Performance of descriptors generated with the AGBT framework on eight datasets.**

| Datasets | LD50 | IGC50 | LC50 | LC50DM | LogP | FreeSolv | Lipophilicity | BBBP |
|---|---|---|---|---|---|---|---|---|
| Metric | $R^2$ | $R^2$ | $R^2$ | $R^2$ | $R^2$ | RMSE | RMSE | AUC-ROC |
| AG-FP | 0.647 | 0.788 | 0.713 | 0.75 | 0.838 | 1.018 | 0.664 | 0.677 |
| BT-FP | 0.667 | 0.839 | **0.783**[d] | 0.763 | 0.895 | 1.125 | 0.626 | 0.736 |
| $BT_s$-FP | 0.617 | 0.798 | 0.75 | 0.829 | 0.903 | 1.036 | **0.57**[a] | **0.763**[b] |
| AGBT-FP | **0.671**[d] | **0.842**[d] | 0.776 | 0.781 | 0.885 | **0.994**[c] | 0.663 | 0.738 |
| $AGBT_s$-FP | 0.612 | 0.805 | 0.75 | **0.83**[d] | **0.905**[d] | 1.039 | 0.579 | 0.761 |

A complete result with multiple evaluation metrics is available in Supplementary Table 7; Best performances are produced on [a]GBDT, [b]RF, [c]STDNN, and [d]MTDNN, and are bolded.

The above discussion indicates that SSL can acquire general molecular information and universal molecular descriptors without the guidance of labels. In downstream tasks, the MT-DNN model can also help to extract the task-specific information from related data. As for small datasets, such as the LC50DM dataset (300 samples), the subsequent fine-tuning with an SL strategy is much more promising.

The results of all eight datasets using AG-FP, BT-FP/$BT_s$-FP, and AGBT-FP/$AGBT_s$-FP are shown in Table 2. The fused descriptors (AGBT-FP/$AGBT_s$-FP) achieved the best performance in 5/8 of the tasks. For the LC50 dataset, the AGBT-FP prediction of 0.776 is very close to the best performance of 0.783 obtained by BT-FPs. For Lipophilicity dataset, the performance of $AGBT_s$-FP is RMSE = 0.579. It is close to the best RMSE (0.57). And for the BBBP dataset, the classification performance of $AGBT_s$-FP is AUC-ROC = 0.761, which is almost the same as the best 0.763. The complete results for all 8 datasets with multiple evaluation metrics are shown in Supplementary Table 7.

**Molecular representations and structural genes**. In chemistry, the properties of molecules, such as toxicity, are often determined by some specific functional groups or fragments. Similar to biological genes, molecules have some determinants of their properties, which are called structural genes in this work. For some path-based fingerprints, such as FP2, a molecule is represented by 256 length vectors, each corresponding to a specific fragment. However, it is difficult to achieve the best results from such a fingerprint, as shown in Fig. 2a, b. The proposed AGBT-FP is a 512-dimensional fingerprint, with each dimension being a projection of various physical information about the molecule. In this section, we hope to characterize the key dimensions of AGBT-FPs to identify the structural genes.

Using a random forest algorithm, we performed a feature importance analysis of AGBT-FPs. As shown in Supplementary Fig. 13, for the LD50, IGC50, and LC50 datasets, the top three features in the feature importance ranking are all from algebraic graph-based descriptors. For the toxicity datasets, the ratio of components from AG-FPs and BT-FPs in the AGBT-FPs is 188: 324, as shown in Fig. 2g and Supplementary Fig. 13. For the LC50DM dataset, the most important feature is from BT-FPs and the 2nd and 3rd important features are from AG-FPs. This implies that the multiscale weighted colored algebraic graph-based molecular descriptors contribute the critical molecular features, which are derived from embedding specific physical and chemical information into graph invariants. The top three important features of the LD50 set are illustrated in Fig. 2e, where each point represents a molecule and the toxicity is represented by the color. It is easy to find that the top three important dimensions in AGBT-FP, denoted as Feature 1, Feature 2, and Feature 3, divide the molecules into two groups: one can be distinguished by Feature 3 and the other is a linear combination of Feature 1 and Feature 2. This means the molecule can be classified by just three key dimensions (features), indicating that

these three features, or structural genes, dominate the intrinsic characteristics of molecules. However, since predicting molecular toxicity is complex, it is difficult to directly distinguish the toxicity of each molecule in AGBT-FPs through the first three dimensions. Similarly, the visualizations for the IGC50, LC50, and LC50DM datasets can be seen in Supplementary Fig. 14.

We projected both AGBT-FPs and $AGBT_s$-FPs into an orthogonal subspace by principal component analysis. As shown in Fig. 2h, the first two principal components of AGBT-FPs can roughly divide the data into two clusters and the molecules in the same cluster have similar toxicity. Similarly, the top two components of $AGBT_s$-FPs are given in Fig. 2h. Along the direction of the first principal components, the molecular data can be well clustered according to the toxicity, with low toxic molecules on the left (green) and higher toxic molecules on the right (red). It indicates these two molecular fingerprints contain very different information. As shown in Supplementary Fig. 15, for AGBT-FPs we need 112 components to explain 90% of the variance, while for $AGBT_s$-FPs we only need 48 components. The top two principal components of AGBT-FPs are just explaining 9% and 8% of the variance, which indicates that, since there is no labeled data to train the model, the generated AGBT-FPs represent general information about the molecular constitution rather than specific molecular properties. The first two components for $AGBT_s$-FPs can explain 40% and 13% of the variance respectively, which indicates that by using SL-based fine-tuning training, the model can effectively capture task-specific information.

The $AGBT_s$-FP model performs better in predicting specific properties because the labeled data are used to train the model during fine-tuning. It should be noted that some molecular information irrelevant to that particular property might be lost in this way. This strategy leads to better results for some datasets with minimal data, such as LC50DM, whose small amount of data is not enough to effectively obtain property-specific information in downstream tasks. However, if more downstream data are available, such as LD50, IGC50, and LC50, downstream machine learning methods can also derive property-specific information from general molecular information. For example, AGBT-FPs perform better on LD50, IGC50, and LC50 datasets.

Despite many efforts in the past decade, accurate and reliable prediction of numerous molecular properties remains a challenge. Recently, deep bidirectional transformers have become a popular approach in molecular science for their ability to extract fundamental constitutional information of molecules from massive SSL. However, they neglect crucial stereochemical information. The algebraic graph is effective in simplifying molecular structural complexity but relies on the availability of 3-D structures. We propose an AGBT framework for molecular property prediction. Specifically, element-specific multiscale weighted colored algebraic subgraphs are introduced to characterize crucial physical/chemical interactions. Moreover, for small datasets, we introduce a supervised fine-tuning procedure

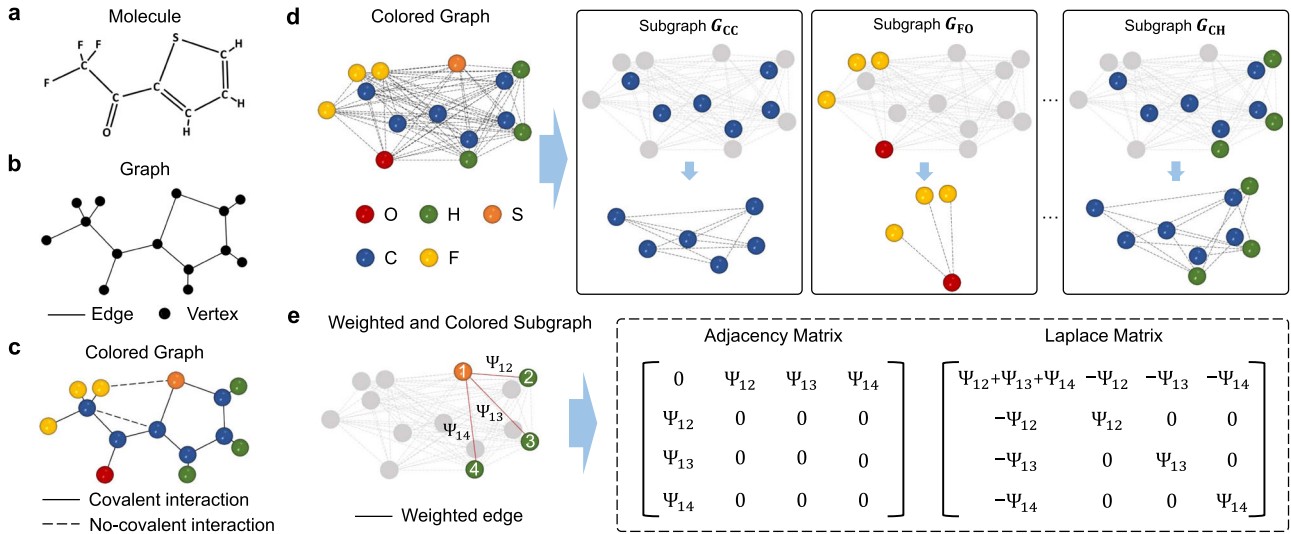

**Fig. 3 Illustration of weighted colored element-specific algebraic graphs. a** The molecular structure of 2-Trifluoroacetyl. **b, c** Represent a traditional graph representation and a colored graph representation, respectively. **d** Illustration of the process of decomposing a colored graph into element-specific CC, FO, and CH subgroups, where element refers to the chemical element in this study, e.g., H, C, N. **e** Illustration of weighted colored element-specific subgraph $G_{SH}$, its adjacency matrix, and Laplacian matrix, where $\Psi$ refers to the weight of the edge in subgraph.

to the standard pre-trained SSL to focus on task-specific information. These approaches are paired with random forest, gradient boosted decision trees, multitask deep learning, and deep neural network algorithms in AGBT. We demonstrate that the proposed AGBT framework achieves $R^2$ values of 0.671, 0.842, 0.783, 0.830, and 0.905 on LD50, IGC50, LC50, LC50DM, and FDA log$P$ dataset, respectively. In the datasets FreeSolv and Lipophilicity, we obtained RMSE scores of 0.994 and 0.579, respectively, and in the classification dataset BBBP, we obtained an AUC-ROC score of 0.763. Our model can be easily extended to the prediction of other molecular properties. Our results show that the proposed AGBT is a robust, and powerful framework for studying various properties of small molecules in drug discovery and environmental sciences.

## Methods
**Algebraic graph-based molecular fingerprints (AG-FPs).** Graph theory can encode the molecular structures from a high-dimensional space into a low-dimensional representation. The connections between atoms in a molecule can be represented by graph theory, as shown in Fig. 3a, b. However, ignoring the quantitative distances between atoms and the different atomic types in traditional graphs will result in the loss of critical chemical and physical information about the molecule. Element-specific multiscale weighted colored graph representations can quantitatively capture the patterns of different chemical aspects, such as van der Waals interactions and hydrogen bonds between different atoms[24]. Figure 3c illustrates a colored graph representation, which captures the element information by using colored vertices and different edges are corresponding to different pair-wise interactions in the molecule. Moreover, the algebraic graph features are easily obtained from the statistics of the eigenvalues of appropriated graph Laplacians and/or adjacency matrices[24].

As shown in Fig. 3d, for a given molecule, we first construct element-specific colored subgraphs using selected subsets of atomic coordinates as vertices,

$$\mathcal{V} = \{(\mathbf{r}_i, \alpha_i) | \mathbf{r}_i \in \mathbb{R}^3; \alpha_i \in \mathcal{E}; i = 1, 2, ..., N\} \quad (1)$$

where $\mathcal{E} = \{$ H, C, N, O, S, P, F, Cl, Br, ... $\}$ is a set of commonly occurring element types for a given dataset. And $i$th atom in a $N$-atom subset is labeled both by its element type $\alpha_i$ and its position $\mathbf{r}_i$. We denote all the pairwise interactions between element types $\mathcal{E}_{k_1}$ and $\mathcal{E}_{k_2}$ in a molecule by fast-decay radial basis functions

$$\mathcal{W} = \left\{ \Psi(\|\mathbf{r}_i - \mathbf{r}_j\|; \eta_{k_1 k_2}) | \alpha_i = \mathcal{E}_{k_1}, \alpha_j = \mathcal{E}_{k_2}; i, j = 1, 2, ..., N; \right.$$
$$\left. \times \|\mathbf{r}_i - \mathbf{r}_j\| > r_i + r_j + \sigma \right\} \quad (2)$$

where $\|\mathbf{r}_i - \mathbf{r}_j\|$ is the Euclidean distance between $i$th and $j$th atoms in a molecule, $r_i$ and $r_j$ are the atomic radii of $i$th and $j$th atoms, respectively, and $\sigma$ is the mean standard deviation of $r_i$ and $r_j$ in the dataset. Figure 3e gives the illustration of Laplace and adjacency matrices based on the weighted colored subgraph. For the

prediction of toxicity, van der Waals interactions are much more critical than covalent interactions and thus the distance constraint ($\|\mathbf{r}_i - \mathbf{r}_j\| > r_i + r_j + \sigma$) is used to exclude covalent interactions. In biomolecules, we usually choose generalized exponential functions or generalized Lorentz functions as $\Psi$, which are weights between graph edges[49]. Here, $\eta_{k_1 k_2}$ in the function is a characteristic distance between the atoms and thus is a scale parameter. Therefore, we generate a weighted colored subgraph $G(\mathcal{V}, \mathcal{W})$. In order to construct element-specific molecular descriptors, the multiscale weighted colored subgraph rigidity is defined as

$$RI^G(\eta_{k_1 k_2}) = \sum_i \mu_i^G(\eta_{k_1 k_2}) = \sum_i \sum_j \Psi(\|\mathbf{r}_i - \mathbf{r}_j\|; \eta_{k_1 k_2}),$$
$$\alpha_i = \mathcal{E}_{k_1}, \alpha_j = \mathcal{E}_{k_2}; \|\mathbf{r}_i - \mathbf{r}_j\| > r_i + r_j + \sigma \quad (3)$$

where $\mu_i^G(\eta_{k_1 k_2})$ is a geometric subgraph centrality for the $i$th atom[50]. The $\mu_i^G(\eta_{k_1 k_2})$ here is a weight subgraph generalization of Gaussian network model or a subgraph generalization of the multiscale flexibility-rigidity index. The summation over $\sum_j \mu_i^G(\eta_{k_1 k_2})$ represents the total interaction strength for the selected pair of element types $\mathcal{E}_{k_1}$ and $\mathcal{E}_{k_2}$, which provide the element-specific coarse-grained description of molecular properties. By choosing appropriate element combinations $k_1$ and $k_2$, the characteristic distance $\eta_{k_1 k_2}$, and subgraph weight $\Psi$, we finally construct a family of element-specific, scalable (i.e., molecular size independent), multiscale geometric graph-based molecular descriptors[24].

To generate associated algebraic graph fingerprints, we construct corresponding graph Laplacians and/or adjacency matrices. For a given subgraph, its matrix representation can provide a straightforward description of the interaction between subgraph elements. To construct a Laplacian matrix, we consider a subgraph $G_{k_1 k_2}$ for each pair of element types $\mathcal{E}_{k_1}$ and $\mathcal{E}_{k_2}$ and define an element-specific weighted colored Laplacian matrix $L(\eta_{k_1 k_2})$ as[24]

$$L_{ij}(\eta_{k_1 k_2}) = \begin{cases} -\Psi(\|\mathbf{r}_i - \mathbf{r}_j\|) & \text{if } i \neq j, \alpha_i = \mathcal{E}_{k_1}, \alpha_j = \mathcal{E}_{k_2} \text{ and } \|\mathbf{r}_i - \mathbf{r}_j\| > r_i + r_j + \sigma; \\ -\sum_j L_{ij} & \text{if } i = j \end{cases} \quad (4)$$

Mathematically, the element-specific weighted Laplacian matrix is symmetric, diagonally dominant, and positive semi-definite, and thus all the eigenvalues are non-negative. The first eigenvalue of the Laplacian matrix is zero because the summation of every row or every column of the matrix is zero. The first non-zero eigenvalue of $L_{ij}(\eta_{k_1 k_2})$ is the algebraic connectivity (i.e., Fiedler value). Furthermore, the rank of the zero-dimensional topological invariant, which represents the number of the connected components in the graph, is equal to the number of zero eigenvalues of $L_{ij}(\eta_{k_1 k_2})$. A certain connection between geometric graph formulation and algebraic graph matrix can be defined by:

$$RI^g(\eta_{k_1 k_2}) = \text{Tr} \, L(\eta_{k_1 k_2}), \quad (5)$$

where Tr is the trace. Therefore, we can directly construct a set of element-specific weighted colored Laplacian matrix-based molecular descriptors by the statistics of nontrivial eigenvalues $\{\lambda_i^L\}_{i=1,2,3,...}$, i.e., summation, minimum, maximum, average,

and standard deviation of nontrivial eigenvalues. Note that the Fiedler value is included as the minimum.

Similarly, an element-specific weighted adjacency matrix can be defined by

$$A_{ij}(\eta_{k_1 k_2}) = \begin{cases} \Psi(\|\mathbf{r}_i - \mathbf{r}_j\|) & \text{if } i \neq j, \alpha_i = \mathcal{E}_{k_1}, \alpha_j = \mathcal{E}_{k_2} \text{ and } \|\mathbf{r}_i - \mathbf{r}_j\| > r_i + r_j + \sigma; \\ 0 & \text{if } i = j \end{cases} \quad (6)$$

Mathematically, adjacency matrix $A_{ij}(\eta_{k_1 k_2})$ is symmetrical non-negative matrix. The spectrum of the proposed element-specific weighted colored adjacency matrix is real. A set of element-specific weighted labeled adjacency matrix-based molecular descriptors can be obtained by the statistics of $\{\lambda_i^A\}_{i=1,2,3,...}$, i.e., summation, minimum, maximum, average, and standard deviation of all positive eigenvalues. To predict the properties of a molecule, graph invariants, such as the eigenvalue statistics of the above matrix, can capture topological and physical information about the molecules, which is named AG-FPs. Detailed parameters of the proposed algebraic graph model can be found in Supplementary Note 3.

**Bidirectional transformer fingerprints (BT-FPs)**. Unlike RNN-based models, DBT is based on the attention mechanism and it is more parallelable to reduce the training time with massive data[28]. Based on the DBT architecture, Devlin et al.[27] introduced a representation model called BERT for natural language processing. There are two tasks involving BERT, masked language learning, and consecutive sentences classification. Masked language learning uses a partially masked sentence (i.e., words) as input and employs other words to predict the masked words. The consecutive sentences classification is to classify if two sentences are consecutive. In the present work, the inputs of the deep bidirectional transformer are molecular SMILES strings. Unlike the sentences in traditional BERT for natural language processing, the SMILES strings of different molecules are not logically connected. However, we train the bidirectional encoder from the transformer to recover the masked atoms or functional groups.

Because a molecule could have multiple SMILES representations, we first convert all the input data into canonical SMILES strings, which provide a unique representation of each molecular structure[51]. Then, a SMILES string is split into symbols, e.g., C, H, N, O, =, Br, etc., which generally represent the atoms, chemical bonds, and connectivity, see Supplementary Table 2 for more detail. In the pre-training stage, we first select a certain percentage of the input symbols randomly for three types of operations: mask, random changing, and no changing. The purpose of the pre-training is to learn fundamental constitutional principles of molecules in a SSL manner with massive unlabeled data. A loss function is built to improve the rate of correctly predicted masked symbols during the training. For each SMILES string, we add two special symbols, <s> and <\s>. Here, <s> means the beginning of a SMILES string and <\s> is a special terminating symbol. All symbols are embedded into input data of a fixed length. A position embedding is added to every symbol to indicate the order of the symbol. The embedded SMILES strings are fed into the BERT framework for further operation. Supplementary Fig. 2 shows the detailed process of pre-training procedure. In our work, more than 1.9 million unlabeled SMILES data from CheMBL[32] are used for the pre-training so that the model learns basic "syntactic information" about SMILES strings and captures global information of molecules.

Both BT-FPs and BT_s-FPs are created in the fine-tuning training step, which further learns the characteristics of task-specific data. Two types of fine-tuning procedures are used in our task-specific fine-tuning. The first type is still based on the SSL strategy, where the task-specific SMILES strings are used as the training inputs, as shown in Supplementary Fig. 3. To accurately identify these task-specific data, only the "mask" and "no changing" operations are allowed in this fine-tuning. The resulting latent-space representations are called BT-FPs.

The second fine-tuning procedure is based on an SL strategy with labeled task-specific data. As shown in Supplementary Fig. 4, when dealing with multiple datasets with cross-dataset correlations, such as four toxicity datasets in the present study (Supplementary Table 4), we make use of all the labels of four datasets to tune the model weights via supervised learning before generating the latent-space representations (i.e., BT_s-FPs), which significantly strengthens the predictive power of the model on the smallest dataset.

In our DBT, an input SMILES string has a maximal allowed length of 256 symbols. During the training, each of the 256 symbols is embedded into a 512-dimensional vector that contains the information of the whole SMILES string. In this extended $256 \times 512$ representation, one can, in principle, select one or multiple 512-dimensional vectors to represent the original molecule. In our work, we choose the corresponding vector of the leading symbol <s> of a molecular SMILES string as the bidirectional transformer fingerprints (BT-FPs or BT_s-FPs) of the molecule. In the downstream tasks, BT-FPs or BT_s-FPs are used for molecular property prediction. Detailed model parameters can be found in Supplementary Note 3.

**Reporting summary**. Further information on research design is available in the Nature Research Reporting Summary linked to this article.

## Data availability

The pre-training dataset used in this work is CheMBL26, which is available at https://ftp.ebi.ac.uk/pub/databases/chembl/ChEMBLdb/releases/chembl_26/. To ensure the reproducibility of this work, the eight datasets used in this work, including four quantitative toxicity datasets (LD50, IGC50, LC50, and LC50DM), partition coefficient dataset, FreeSolv dataset, Lipophilicity dataset, and BBBP dataset, are available at https://weilab.math.msu.edu/Database/.

## Code availability

The overall models and related code have been released as an open-source code and is also available in the Github repository: https://github.com/ChenDdon/AGBTcode[52].

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

## Acknowledgements

The work of Dong Chen, Xin Chen, Yi Jiang and Feng Pan was supported in part by the National Key R&D Program of China (2016YFB0700600). The work of Kaifu Gao and Guo-Wei Wei was supported in part by NSF grants DMS-2052983, DMS1761320, IIS1900473, NIH grants GM126189, and GM129004, Bristol-Myers Squibb, and Pfizer. The work of Duc Nguyen was supported in part by NSF grant DMS-2053284 and University of Kentucky start-up fund. The work of Dong Chen was also partly supported by Michigan State University.

## Author contributions

Dong Chen designed the project, performed computational studies, analyzed data, wrote the first draft, and revised the manuscript. Kaifu Gao, Xin Chen, and Yi Jiang analyzed data, Duc Duy Nguyen analyzed data and revised the manuscript. Guo-Wei Wei conceptualized and supervised the project, revised the manuscript, and acquired funding. Feng Pan conceptualized and supervised the project and acquired funding.

## Competing interests

The authors declare no competing interests.
