## [Peer Review File · Nature Communications]

Reviewers' Comments:

Reviewer #1:

Remarks to the Author:

Chen et al. present an interesting study investigating the use of AGBT encoding of molecules with machine learning algorithms for molecular property prediction. The strength of this method is shown in Table 1 where it outperforms a range of published methods on three molecular property prediction tasks.

The authors should consider updating the title from Molecular Prediction to Molecular Property Prediction.

I am missing a table specifying the pre-training, training and test set sizes for the five considered tasks. That is, those tasks shown in Tables 1 and 2. Also, these tables should state if any of the published methods using subsets of these datasets for any reason, as it is important to compare performance using the same datasets. At least some task seem structure-based and for those the PDB id of the protein must be stated in this new table too.

Where and how are the hyperparameters of the algorithms in Figures 2c and 2d being tuned? Which algorithm was used for Figures 2a and 2b? How exactly pre-training was carried out and why it was beneficial in practice. This should be clarified in the figure too.

R2 refers to the coefficient of determination or the square of Pearson's correlation? These provide in general different values, the exception being when linear regression is employed.

Introducing self-supervising learning by comparison to unsupervised learning is advised.

PCA plots are shown in Figure 2: the % of variance captured by the first two PCs should be stated.

The five datasets (pages 4-5) should be better motivated. How Daphnia Magna and Fathead Minnows are employed in preclinical drug discovery? Which are the two solvents considered for log P and what is its relevance for drug design? [38] is from 2003, adding a more reference and indicating if attrition rates due to toxicity and side effects are still the same today is advisable.

I appreciate the explanation of the employed encoding in Figure 5. This is very helpful to understand what has been done. However, I think that the terminology should be explicitly explained for clarity. For example: element-specific refers to the chemical element? What multi-scale refers to? Psi seems inter-atomic distances, please state so in Figure 5. Adjacency and Laplace are matrices processed to provide a feature vector for each data instance, but how large is this feature vector for each task? It is not clear either how their attention mechanism works.

The authors are well-known for their excellent work on structure-based binding affinity prediction using PDBbind benchmarks. They should apply their new method to one of the benchmarks and compare all the published methods (the results of the best of these methods are compiled at <https://doi.org/10.1002/wcms.1465>).

The authors provide their code for reproducibility and use on other training and test sets. As far as I can see, the pre-trained models are downloaded and the code to pretrain these models is not provided, so it is not clear how this method can be pre-train with other datasets.

To sum up, I think this is an important contribution from the authors, which just requires minor revisions to improve clarity and benchmarking. The fact that outperforms all other methods in 5 different tasks is its strength.

Reviewer #2:

Remarks to the Author:

Major comments:

1. In the abstract, the authors claim "Although some machine learning models, such as bidirectional encoder from transformer, can incorporate massive unlabeled molecular data into molecular representations via a self-supervised learning strategy, it neglects three-dimensional (3D) stereochemical information". This claim is confirmed neither by this paper's results nor by citing other articles. There is no proof that adding Algebraic Graph representation improves overusing only bidirectional transformer embeddings - in Table 1 and Figure S9, the improvement of AGBT-FP over BT-FP looks minor. Also, no comparison AGBT-FP vs. AG-FP was provided. It would be great to perform several additional experiments to prove these statements:

a. Provide additional results in Table 1 and Figure 2 for a model that uses either only AG representation (AG-FP) or BT representation (BT-FP) on every dataset.

b. Provide an additional experimental setup on 3d molecular properties prediction task, for example, dipole moment, and show the performance gap with and without AG representation.

2. The authors conducted the experiment on an extremely small dataset IGC50 and mentioned the overfitting issue possibility "The small size leads to difficulties in building a robust prediction model." (page 4, lines 117-118). Although they claim "It suggests that our AGDMT framework not only can overcome overfitting problem but also is not sensitive to dataset size" (page 4, lines 129-130), they did not provide any statistical proof of this claim. It would be great to add mean and standard deviations in Table 1 and Figure 2 to show the results' significance.

Minor Comments:

The authors propose a scheme to select optimal fingerprints with the random forest algorithm. It would be great to provide the analysis of selected optimal fingerprints in the main text - move Figure S10 from supplementary, provide a number of fingerprints from AG-FP and some fingerprints from BT-FP, along with an interpretation of the most important fingerprints.

Reviewer #3:

Remarks to the Author:

In this study, the authors combined algebraic graphs and bidirectional encoders to produce a latent space which can be used in machine learning tasks for property predictions. In previous works, deep learning methods to capture 'learn' features/latent variables from chemical structures (e.g. encoders using chemical SMILES) lacked the ability to capture important 3D information such as stereochemistry and van der Waals interactions. The authors solve this by using color algebraic graphs combined with bidirectional transformers. This improvement is novel. However, this work has several critical issues, especially for the design of benchmark study, that need to be addressed.

Major issues:

The authors should clearly understand that the major shortcoming of machine learning studies of small molecules is the data, not the descriptors. And so far the QSAR modeling is limited by the availability of the training data and also cannot solve "activity cliff" issue. The research reported in this study did not solve these two relevant challenges but provide a new type of descriptors. This must be clearly stated in the introduction and final discussion sections.

The benchmark study is confusing and not properly designed. First, only using R2 as the criterion to compare different models is not sufficient. Second, only using five datasets for the benchmark study is too limited compared to hundreds or even thousands of toxicity/property datasets publicly available. There have been so many benchmark studies published previously to compare different descriptors and modeling approaches. So far there is no specific one can be better than all others by modeling all types of datasets. This work is not an exception. Therefore, by using so few datasets (and indicating the new descriptor is the best in all models), "Cherry Picking" is the first impression of this work. Third, all the datasets were divided into training/test sets, which is not proper to show the performance of resulted models. Cross validation has been shown to be a better way compared to arbitrarily divided training/test sets.

Classification models were not included in this study so the applicability of the new descriptors is still doubtful.

For a better comparison, the authors must train machine learning models using different descriptor packages for the same datasets. This will be a much better comparison (e.g., using the same methods and same training/test set splits) between their fingerprints and other descriptors compared to the current very obscure descriptions of other people's works (only references were provided but no details at all).

Minor issues:

Table 1 is also confusing, as ML methods are mixed with descriptors as well as (presumably) different datasets (e.g., FDA). Also, this table only has three of the toxicity/property datasets listed. Why were the others left out?

Are there any correlations that can be made from the latent space to physicochemical properties? Or is the latent space and entirely black box? For example, can the clusters identified by the PCA on the latent variables be extracted for any information about molecular properties? If not, I suggest the authors announce this pitfall as compared to other descriptors whose relevance can more readily be more chemically and toxicologically interpretable.

Dear Editor and Reviewers:

We would like to thank the reviewers for their valuable comments on our manuscript entitled: "Algebraic Graph-assisted Bidirectional Transformers for Molecular Prediction" (Manuscript No.: NCOMMS-21-02805). These comments have guided us to improve our manuscript. We have carefully gone through referees' comments, and have revised the manuscript accordingly. The responses are as following:

Reviewer 1:

Comment:

Chen et al. present an interesting study investigating the use of AGBT encoding of molecules with machine learning algorithms for molecular property prediction. The strength of this method is shown in Table 1 where it outperforms a range of published methods on three molecular property prediction tasks.

1. The authors should consider updating the title from Molecular Prediction to Molecular Property Prediction.

Answer: Thank you for the suggestion. The title has been modified.

2. I am missing a table specifying the pre-training, training and test set sizes for the five considered tasks. That is, those tasks shown in Tables 1 and 2. Also, these tables should state if any of the published methods using subsets of these datasets for any reason, as it is important to compare performance using the same datasets. At least some task seem structure-based and for those the PDB id of the protein must be stated in this new table too.

Answer: Thank you for your comment and reminder. In the revised Supporting Information, a summary of all datasets is listed in Table S1. We provide a detailed description of all the datasets used in this work, including the training set size, the validation size, the test set size, and the methods of data splitting. We have added 3 additional datasets for further analysis in the revised version. All the data used in this study and the data splitting methods are the same as those in the literature. The corresponding references are also given in the datasets section of Supporting Information.

3. Where and how are the hyperparameters of the algorithms in Figures 2c and 2d being tuned? Which algorithm was used for Figures 2a and 2b? How exactly pre-training was carried out and why it was beneficial in practice. This should be clarified in the figure too.

Answer: The 'AGBT model parametrization' section in the Supporting Information records the settings of all hyperparameters for the models in this paper. In the 'SSL-based and SL-based fine-tuning' subsection, we provide the hyperparameters used in the fine-tuning stage. In Figures 2c, the red bars correspond to BT-FPs' result, which is derived from the model fine-tuned by self-supervised Learning (SSL) algorithm. And the yellow bars correspond to the AGBT-FP's result, and this fingerprint comes from the fusion of AG-FPs and BT-FPs. In Figure 2d, AGBT-FP and AGBT_s-FPs are applied to the SSL finetuning and SL fine-tuning, respectively. For Figures 2a and 2b, the best performance is derived from multitask deep neural network machine learning algorithm. In the revised version, we provide results in Table 2 and the complete result in Table S6. In the Section of 'Bidirectional transformer fingerprints (BT-FPs)', we introduce the pre-training procedure. The detailed mechanism of pre-training has given in the 'Bidirectional transformer model parametrization' section in the Supporting Information, and Figure S2 gives the whole structure of the model used in pre-training. With the pre-training method, we can train the model using a large amount of unlabeled data, which can allow the model to acquire more information. Subsequent fine-tuning then allows the model to handle specific tasks.

4. R2 refers to the coefficient of determination or the square of Pearson's correlation? These provide in general different values, the exception being when linear regression is employed.

Answer: In this work, R^2 refers to the squared Pearson correlation coefficient, and we added the formula in the 'Evaluation Metrics' section of the Supporting Information.

5. Introducing self-supervising learning by comparison to unsupervised learning is advised.

Answer: Thank you for the suggestion. We add the comparison in the 'Introduction' (Page2). And the detailed algorithm of SSL is provided in the 'Method' section and the 'SSL-based pre-training' section of Supporting Information.

6. PCA plots are shown in Figure 2: the % of variance captured by the first two PCs should be stated.

Answer: Thanks for the suggestion. We modified the PCA plots in the Figure 2.

7. The five datasets (pages 4-5) should be better motivated. How Daphnia Magna and Fathead Minnows are employed in preclinical drug discovery? Which are the two solvents considered for log P and what is its relevance for drug design? [38] is from 2003, adding a more reference and indicating if attrition rates due to toxicity and side effects are still the same today is advisable.

Answer: Thanks for the suggestion. The detailed descriptions of datasets are provided in the 'Datasets' section of the Supporting Information, including additional three datasets, e.g., FreeSolv, Lipophilicity, and BBBP. For toxicity datasets, as described in 'Datasets' section of the Supporting Information "LC50DM, which represents the concentration of test37chemicals in the water in milligrams per liter that cause 50% Daphnia Magna to die after 48h. LC50 and LC50DM were original from <http://cfpub.epa.gov/ecotox/>". For the logP set, the octanol and water were considered for partition coefficients prediction, which is important for drug discovery. To better validate the performance of our model, all datasets used in this work were the same as those in the published works.

8. I appreciate the explanation of the employed encoding in Figure 5. This is very helpful to understand what has been done. However, I think that the terminology should be explicitly explained for clarity. For example: element-specific refers to the chemical element? What multi-scale refers to? Psi seems inter-atomic distances, please state so in Figure 5. Adjacency and Laplace are matrices processed to provide a feature vector for each data instance, but how large is this feature vector for each task? It is not clear either how their attention mechanism works.

Answer: Thanks for the comments. The element in 'element specific' refers to the common occurred chemical elements in the datasets, e.g., H, C, N, O, S, P, F, Cl, Br. The kernel functions (Φ) define a continuous multi-scale rigidity function by using the fitting coefficients from the minimization process as shown in Equation 3. The $\mu_i^G(\eta_{k_1 k_2})$ is actually a weight subgraph generalization of Gaussian network model (GNM). Therefore, $\mu_i^G(\eta_{k_1 k_2})$ can be used to represent atomic properties, such as protein B-factors. The weight of the edge in the graph, denoted as Φ , is represented by generalized exponential functions or Lorentz functions in this work, which are derived from non-covalent distances. Statistical values such as maximum, minimum, mean, sum, median, standard deviation, and variance of feature values are used as features of each subgraph to generate a total of 900 features of a molecule when using a specific type of kernel function and matrix. In this work, the two types of kernels, generalized exponential functions or Lorentz functions, will be combined based on the results of cross-validation to finally generate an 1800-dimensional feature to describe the molecule. We have added more explanation in Figure 3, the 'Methods' section of the main script, and the 'Algebraic graph model parametrization' in the Supporting Information.

9. The authors are well-known for their excellent work on structure-based binding affinity prediction using PDBbind benchmarks. They should apply their new method to one of the benchmarks and compare all the published methods (the results of the best of these methods are at <https://doi.org/10.1002/wcms.1465>).

Answer: Thank you for your suggestion. In this study, we focused on the prediction of properties of small

molecules. However, in PDBbind, the labels, binding affinity, are derived from the interactions of proteins and small molecules. Therefore, it is not suitable to use the Transformer method described in this study (but the algebraic graph method can be added to PDBbind). But we added 3 more datasets derived from Moleculenet to validate our work. In addition, we add the discussion in the 'Introduction' (Page2).

10. The authors provide their code for reproducibility and use on other training and test sets. As far as I can see, the pre-trained models are downloaded and the code to pretrain these models is not provided, so it is not clear how this method can be pre-train with other datasets.

Answer: Our latest code is released in Github at <https://github.com/ChenDdon/AGBTcode>, which includes pre-training parameters. The detailed parametrization process is also provided in the 'Bidirectional transformer model parametrization' section of the Supporting Information.

11. To sum up, I think this is an important contribution from the authors, which just requires minor revisions to improve clarity and benchmarking. The fact that outperforms all other methods in 5 different tasks is its strength.

Answer: Thank your for the comments and the suggestions.

Reviewer 2:

Major comments:

1. In the abstract, the authors claim "Although some machine learning models, such as bidirectional encoder from transformer, can incorporate massive unlabeled molecular data into molecular representations via a self-supervised learning strategy, it neglects three-dimensional (3D) stereochemical information". This claim is confirmed neither by this paper's results nor by citing other articles. There is no proof that adding Algebraic Graph representation improves overusing only bidirectional transformer embeddings - in Table 1 and Figure S9, the improvement of AGBT-FP over BT-FP looks minor. Also, no comparison AGBT-FP vs. AG-FP was provided. It would be great to perform several additional experiments to prove these statements:
 - a. Provide additional results in Table 1 and Figure 2 for a model that uses either only AG representation (AG-FP) or BT representation (BT-FP) on every dataset.
 - b. Provide an additional experimental setup on 3d molecular properties prediction task, for example, dipole moment, and show the performance gap with and without AG representation.

Answer:

Thanks for your comments and suggestions. In this study, the self-supervised learning strategy is based on massive unlabeled molecular data, where the data refers specifically to canonical SMILES strings. However, the simplified molecular-input line-entry system (SMILES) saves the structure as a string while losing some 3D information. For example, the cis-trans isomerism is important in chemistry, as shown in Figure R1a, if two Cl atoms are oriented in opposing directions, the diastereomer is referred to as Trans. If two Cl atoms are oriented in the same directions, as shown in Figure R1b, the diastereomer is referred to as Cis. However, the canonical SMILES strings for these two structures are the same. Since the canonical SMILES string will be utilized as input in the SSL training process as well as in the BT-FP generation. This means that this BF-FP cannot distinguish these two structures. On the contrary, AG-FP based on the algebraic graph theory can distinguish this class of structures by obtaining 3D information through graph invariants (eigenvalues), as shown in Figure R1c and R1d.

a) We have added the results for BT-FPs, AG-FPs, and AGBT-FPs in Table 2. More detailed results with multiple metrics are given in Table S6. The 5 descriptors generated by our method, AG-FP, BT-FP, BTs-FP,

ABGT-FP, AGBTs-FP obtained, 1, 3, 4, 8, 7 best scores on 8 datasets for 23 metrics, respectively. In the FreeSolv dataset, the AG-FP achieves the best result in $R^2(0.935)$. Although BT-FP achieves good results on various tasks, it is not able to solve problems like cis-trans isomerism. In this study, with the help of Algebraic Graph-based features, the Algebraic Graph-assisted Bidirectional Transformers (AGBT) acquires the ability to deal with such problems.

b) Three additional datasets, including two regression datasets and one classification task, are added in the revised version. The detailed results were provided in Table 2 and Table S6. The comparison with the literature is presented in Table 1 and Table S5.

Figure R1: Application of algebraic graph theory methods to the analysis of cis-trans isomers. **a** and **b** Illustration of Trans-1,2-Dichlorocyclohexane and Cis-1,2-Dichlorocyclohexane, these two molecules share the same canonical SMILES. **c** and **d** The trans- and cis- molecular subgraph $G_{Cl,C}$ for the conditions $AG_{E,1.0,1.0}^{Adj}$.

- The authors conducted the experiment on an extremely small dataset IGC50 and mentioned the overfitting issue possibility “The small size leads to difficulties in building a robust prediction model.” (page 4, lines 117-118). Although they claim “It suggests that our AGDMT framework not only can overcome overfitting problem but also is not sensitive to dataset size” (page 4, lines 129-130), they did not provide any statistical proof of this claim. It would be great to add mean and standard deviations in Table 1 and Figure 2 to show the results’ significance.

Answer: Thanks for your suggestion. The detailed results for eight datasets and the standard deviation values for the additional datasets are listed in Table S6 and Table S7, respectively. Our random splitting of datasets into non-overlapped training and test sets can overcome the overfitting problem.

In this work, to better compare the performance among descriptors and eliminate the systematic errors generated by the machine learning models, the consensus of the predicted values from 20 different models (generated with different random seeds) was taken for each molecule. Therefore, we end up with only one predicted value for each dataset.

*For the initial five datasets, all data were pre-split into the training and test sets in the way following the literature. For the FreeSolv and Lipophilicity datasets, which are adopted from MoleculeNet, we **randomly** split these datasets (following the same procedure as in MoleculeNet) 10 times. Thereby, for these datasets, ten values for each metric can be generated. The final standard deviation results are listed in Table S7. As for FreeSolv (643 samples) and Lipophilicity (4200 samples), the standard deviations for metric R^2 are quite close, which means that our descriptors have a good consistence in terms of ranking power. However, for both metric RMSE and metric MAE, the performance is better when the dataset contains more samples. We have added more discussion in the main script (Page5 and Page6).*

Minor comments:

1. The authors propose a scheme to select optimal fingerprints with the random forest algorithm. It would be great to provide the analysis of selected optimal fingerprints in the main text - move Figure S10 from supplementary, provide a number of fingerprints from AG-FP and some fingerprints from BT-FP, along with an interpretation of the most important fingerprints.

Answer: Thank you for the suggestion. The related figures were added to Figure 2g. The additional discussion has also been added to the 'Discussion' section (page8), "For the toxicity datasets, the ratio of components from AG-FPs and BT-FPs in the AGBT-FPs is 188: 324, as shown in Figure 2g and Figure S10. For the LC50DM dataset, the most important feature is from BT-FPs and the 2nd and 3rd important features are from AG-FPs. This implies that the multiscale weighted colored algebraic graph-based molecular descriptors contribute the critical molecular features, which are derived from embedding specific physical and chemical information into graph invariants." And in the 'Algebraic graph-based molecular fingerprints' section (Page 11), we give the explanation of the algebraic graph features.

Reviewer 3:

Comment:

In this study, the authors combined algebraic graphs and bidirectional encoders to produce a latent space which can be used in machine learning tasks for property predictions. In previous works, deep learning methods to capture 'learn' features/latent variables from chemical structures (e.g. encoders using chemical SMILES) lacked the ability to capture important 3D information such as stereochemistry and van der Waals interactions. The authors solve this by using color algebraic graphs combined with bidirectional transformers. This improvement is novel. However, this work has several critical issues, especially for the design of benchmark study, that need to be addressed.

Major issues:

1. The authors should clearly understand that the major shortcoming of machine learning studies of small molecules is the data, not the descriptors. And so far the QSAR modeling is limited by the availability of the training data and also cannot solve "activity cliff" issue. The research reported in this study did not solve these two relevant challenges but provide a new type of descriptors. This must be clearly stated in the introduction and final discussion sections.

Answer: Thanks for the comments. It is true that the performance of machine learning is mainly limited by data size, quality, and the lack of proper labels. However, there are ways to improve the performance of machine learning for a given dataset. For example, there is a huge collection of unlabeled data available nowadays, e.g., ZINC15(0.6 billion) and ChEMBL26(over 1.9 million). In this study, we provide a self-supervised learning method using unlabeled data (ChEMBL26). Using this method, we can improve the performance of machine learning for small samples, e.g., LC50DM with only 353 samples and FreeSolv with only 643 samples. With the help of these unlabeled data, our method can eventually achieve outstanding results, as shown in Table 2, Table S5, and Table S6. An "activity cliff" is often defined as a pair of structurally similar compounds that are active against the same target, but differ significantly in their properties. The distinction of similar structures in terms of descriptors is one way to address this problem. In the present work, we use features of algebraic graph theory that can distinguish similar structures to a large extent. As an example, the cis-trans structure in Figure R1 differs only in the orientation of the Cl elements, and ordinary SMILES cannot distinguish these two structures, while our algebraic graph theory-based approach can distinguish such structures from graph invariants (eigenvalues). This shows that the proposed method is capable of handling such problems. We add descriptions in the Introduction (Page2) and Discussion (Page6).

2. The benchmark study is confusing and not properly designed. First, only using R^2 as the criterion to compare different models is not sufficient. Second, only using five datasets for the benchmark study is too limited compared to hundreds or even thousands of toxicity/property datasets publicly available. There have been so many benchmark studies published previously to compare different descriptors and modeling approaches. So far there is no specific one can be better than all others by modeling all types of datasets. This work is not an exception. Therefore, by using so few datasets (and indicating the new descriptor is the best in all models), "Cherry Picking" is the first impression of this work. Third, all the datasets were divided into training/test sets, which is not proper to show the performance of resulted models. Cross validation has been shown to be a better way compared to arbitrarily divided training/test sets.

Answer: Thank you for your suggestions. First, to validate the performance of our method, we have added three more datasets in the revised version and compared with the published benchmark. The results are listed in Table 1, and Table S5. The aim of this work is to provide a new framework based on the fusion of algebraic graph theory descriptors with bidirectional Transformer descriptors and to show that the framework can exhibit outstanding performance in molecular property prediction. For the downstream machine learning models, we did not over-tune the hyperparameters to obtain the best ones (otherwise we can achieve better performance for many problems, but that is not what we want to do). For all the eight datasets, the same set of hyperparameters is taken for each descriptor in our method. The details of the parameter settings can be found in the 'Downstream machine learning algorithms' section of Supporting Information.

Additionally, more model performance evaluation metrics are used, including the squared Pearson correlation coefficient (R^2), the root mean squared error (RMSE), mean absolute error (MAE), classification accuracy, and AUC-ROC. The results are shown in Table 1, Table 2, Table S5, and Table S6. In the comparison discussion, the evaluation metrics were followed those from the published work. For example, for 4 toxicity datasets, the literature mainly uses R^2 , while for FreeSolv and Lipophilicity datasets, the literature chooses RMSE.

To ensure consistency in the comparison, all data splitting procedures are following the published work. For 4 toxicity and logP datasets, the training set and test set are already divided in the published work. In the comparison, all the methods mentioned in this paper also follow the same data splitting from the literature. As for FreeSolv and Lipophilicity, we follow the random splitting method used in MoleculeNet, i.e., dividing the data into the training set, validation set, and test sets in the ratio of 8:1:1. We set different random seeds and follow the above method 10 times to obtain 10 different data divisions. To eliminate systematic errors in downstream machine learning models and to better compare differences in molecular descriptors, for each machine learning method, we also trained 20 models using each set of data and used the average of these model predictions as the final prediction. For the classification dataset BBBP, we follow the same data splitting method in MoleculeNet, which is scaffold splitting. The dataset was also split into training, validation, and the test set follow the ratio of 8/1/1.

3. Classification models were not included in this study so the applicability of the new descriptors is still doubtful.

Answer: Thanks for the comments. We have added the additional classification task in this study, the results are listed in Table 1, Table 2, and Table S7. In comparison, our method (AUC-ROC=0.763) outperforms the results obtained in the benchmark model (AUC-ROC_{ECFP}=0.729). For a better comparison, all data were divided according to the literature and performance was evaluated following the literature. Also, for all the downstream machine learning models, we did not tune the parameters to obtain the best ones in this study. For all the above-mentioned datasets, the same set of hyperparameters is taken for each descriptor in our model. Otherwise, we could obtain better results by "Cherry Picking" hyperparameters. The details of the parameter settings can be found in Section 'Downstream machine learning algorithms' of Supporting Information.

4. For a better comparison, the authors must train machine learning models using different descriptor

packages for the same datasets. This will be a much better comparison (e.g., using the same methods and same training/test set splits) between their fingerprints and other descriptors compared to the current very obscure descriptions of other people's works (only references were provided but no details at all).

Answer: Thank you for your advice. We have initially followed your suggestion to recreate the performance of other methods in the comparison. However, we soon run into the problem that the descriptor dimensions in the literature differ from each other dramatically. In the present work, we emphasize a robust predictive framework with a fixed dimension of 512 and the same set of machine learning hyperparameters for all datasets. Unfortunate, the same machine learning setting cannot be applied to the descriptors in the literature without reducing their accuracy or performance. Therefore, it is unfair for us to fit or prune the descriptors of other methods into our uniform framework.

To be fair in our comparison, all datasets mentioned in this work are following the same splitting method as the comparison work. Additionally, we choose to trust the best performance reported in the literature for all existing methods.

The goal of this work is to provide a novel framework based on the fusion of algebraic graph theory descriptors with bidirectional Transformer descriptors and to show that this framework can exhibit outstanding performance in molecular property prediction. For a specific dataset, we did not try to select the best machine learning algorithm and the most suitable hyperparameters to optimize our performance. For all the eight datasets, the same set of hyperparameters is taken for all descriptors in our framework. Nevertheless, our results still outperform other methods or descriptors on most of the data sets.

Minor issues:

1. Table 1 is also confusing, as ML methods are mixed with descriptors as well as (presumably) different datasets (e.g., FDA). Also, this table only has three of the toxicity/property datasets listed. Why were the others left out?

Answer: Thanks for your comment. In the revised Table 1, we compare six datasets, and the remaining two datasets (IGC50 and LC50DM) are shown in Figures 2a and 2b. Also, we have added the comparison results for these two datasets in Table S5.

2. Are there any correlations that can be made from the latent space to physicochemical properties? Or is the latent space and entirely black box? For example, can the clusters identified by the PCA on the latent variables be extracted for any information about molecular properties? If not, I suggest the authors announce this pitfall as compared to other descriptors whose relevance can more readily be more chemically and toxicologically interpretable.

Answer: In the last 'Molecular representations and structural genes' section, we discussed the latent space and the molecular representations. Although, the features extracted from latent space cannot be directly associated with specific chemical properties, the transformers model mentioned in this work is a transformation to project chemical information into a high-dimensional space. As for the PCA method, it is actually a transformation as well. After PCA projection, as shown in Figure 2h, using only the two most important components, the toxicity data can be roughly separated from left to right according to the magnitude of toxicity.

Reviewers' Comments:

Reviewer #1:

Remarks to the Author:

The authors have made an excellent job addressing all my comments. The only question I have left is why they think that using element atom types results in more predictive features than using a higher number of more precise atom types per element? For instance, sybyl atom types distinguishing each element by their hybridisation state like those used in Chimera:
<https://www.cgl.ucsf.edu/chimera/docs/UsersGuide/idadm.html>

Reviewer #2:

Remarks to the Author:

The authors mostly addressed most of my questions to the best of their ability. The paper could be further improved by expanding the introduction and explaining the different machine learning approaches in drug discovery and the benefits of using transformers compared to GANs, genetic algorithms, etc. GANs and GAN/RL systems are very popular and applied to the same and many similar tasks since 2016 and there is abundant experimental evidence published in top peer-reviewed journals.

Currently, it is not clear how transformers compare in similar tasks and GANs are generally ignored in the paper while it would be great to see at least some form of comparison.

Reviewer #3:

Remarks to the Author:

The authors revised the manuscript based on what I suggested and there is no further comments.

Dear Editor and Reviewers:

We would like to thank the reviewers for their valuable comments on our manuscript entitled: "Algebraic Graph-assisted Bidirectional Transformers for Molecular Prediction" (Manuscript No.: NCOMMS-21-02805A). These comments have guided us to improve our manuscript. We have carefully gone through referees' comments, and have revised the manuscript accordingly. The responses are as following:

Reviewer 1:

Comment:

1. The authors have made an excellent job addressing all my comments. The only question I have left is why they think that using element atom types results in more predictive features than using a higher number of more precise atom types per element? For instance, sybyl atom types distinguishing each element by their hybridisation state like those used in Chimera: <https://www.cgl.ucsf.edu/chimera/docs/UsersGuide/idadm.html>

Answer: Thank you for the comment. It is true that more precise atom types, such as sybyl atom types, can give more detailed descriptions for a particular molecule. However, a more precise distinction of atom types will lead to a higher dimensionality of the descriptors (fingerprints). When the same scheme is used for feature extraction of all data, some simple molecules will be described by very sparse and high-dimensional vectors. In this work, all downstream datasets are small and the number of these datasets does not exceed 8000. In this case, using sparse and high-dimensional vectors as fingerprints would have a bad impact on the prediction performance. On the other hand, the elemental atom type (element-specific) approach proposed in this work has been able to give molecule-specific descriptions through relatively low-dimensional and dense spaces. Therefore, the element-specific methods will show better performance on these small data sets.

Reviewer 2:

Comment:

1. The authors mostly addressed most of my questions to the best of their ability. The paper could be further improved by expanding the introduction and explaining the different machine learning approaches in drug discovery and the benefits of using transformers compared to GANs, genetic algorithms, etc. GANs and GAN/RL systems are very popular and applied to the same and many similar tasks since 2016 and there is abundant experimental evidence published in top peer-reviewed journals. Currently, it is not clear how transformers compare in similar tasks and GANs are generally ignored in the paper while it would be great to see at least some form of comparison.

Answer: Thanks for your comment and suggestion. A classical generative adversarial network (GAN) includes a generator and discriminator. Given a training set, this technique learns to generate new data with the same statistics as the training set, and GANs have proven useful for unsupervised learning, semi-supervised learning, supervised learning, and reinforcement learning (RL). In chemistry, it is popular to use specific patterns (e.g., reinforcement learning) to train the generator in GANs so that new molecules can be generated according to some specific requirements. In contrast, for transformers, the encoder and decoder are connected through an attention mechanism, where the decoder part of the transformers can be used to generate molecules similar to the generator. However, in this work, we only utilized the encoder part of the transformer, which is involved in the pre-training and fine-tuning phases, and used features from the latent space for molecular prediction. As a flexible model, the pre-trained encoder in this work can be directly connected to the decoder part for further training, so that the function of generating molecules can also be achieved. We have added the related description and reference of GANs in the "Introduction" (Page 2).

Reviewer 3:

Comment:

1. The authors revised the manuscript based on what I suggested and there is no further comments.

Answer: Thanks for the comment.